# Extracting PAC Decision Trees from Black Box Binary Classifiers (Extended Abstract)

**Ana Ozaki**                                                   ANAOZ@UIO.NO
*Universitetet i Oslo, Norway*
*Universitetet i Bergen, Norway*

**Roberto Confalonieri**                        ROBERTO.CONFALONIERI@UNIPD.IT
*Università degli Studi di Padova, Italy*

**Ricardo Guimarães** *Universitetet i Bergen, Norway*

**Anders Imenes** *Universitetet i Bergen, Norway*

**Editors:** Leilani H. Gilpin, Eleonora Giunchiglia, Pascal Hitzler, and Emile van Krieken

## Extended Abstract

Decision trees are inherently interpretable models, making them attractive as surrogate models for explaining black box neural networks. Nonetheless, current approaches for extracting decision trees from trained neural networks do not provide theoretical guarantees regarding the fidelity of the decision trees wrt. the original model.

In (Ozaki et al., 2025), we leveraged the Probably Approximately Correct (PAC) learning framework (Valiant, 1984) to provide formal guarantees about the fidelity of decision trees extracted from neural networks, focusing explicitly on binary classification problems. PAC guarantees enabled us to assert with confidence that the decision tree's predictions closely match the original model under specified conditions.

A common assumption in PAC learning is the *realizability assumption*, which basically says that the target belongs to the hypothesis space. When the target belongs to the hypothesis space, in theory we know there is a hypothesis that is consistent with the training data, that is, the training error is 0. This is a rather strong assumption in practice because the hypothesis is usually not fully consistent with the training set (Mohri et al., 2018, Section 2.3). On the other extreme, the notion of agnostic PAC learning completely removes the realizability assumption but comes with other challenges. In our work, we kept the realizability assumption but we allowed the hypothesis to be inconsistent with the training set. We proved in (Ozaki et al., 2025, Th.8) that one can allow the hypothesis to misclassify $k$ examples in a training set and give a bound on the minimal number of examples needed for PAC learnability based on $k$. We then related this PAC notion with the notion of fidelity, classically studied within the XAI literature, and we provided a probabilistic definition.

We proposed a decision tree learning algorithm within the PAC framework, called TREPAC, specifically targeting binary classification tasks to ensure theoretical guarantees of approximation accuracy. TREPAC extracts decision trees from trained binary classifiers, which are seen as oracles. TREPAC can thus be considered a model-agnostic post-hoc explanation algorithm. It differs from conventional inductive learning algorithms (such as CART and C4.5) and post-hoc explanation algorithms (such as Trepan) because it uses the

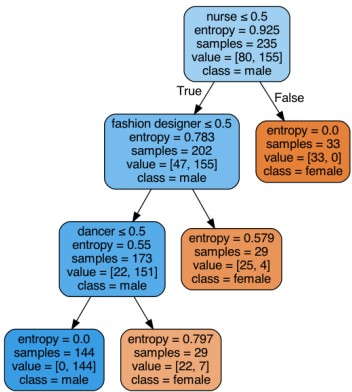

Figure 1: Surrogate tree for RoBERTa-base unveiling occupational bias.

PAC framework to estimate the amount of training data and internal nodes in the resulting decision tree. It uses a membership oracle to classify examples used for training. The PAC framework helped determine how closely the surrogate decision tree replicates the neural network's behavior, particularly critical when addressing sensitive predictions.

In our experiments, we extracted PAC-guaranteed decision trees from BERT-based language models trained to predict occupational gender association, thus revealing and quantifying inherent biases. The decision trees provide explicit rules, clarifying which occupations are most impacted by gender biases learned by BERT. Fig. 1 shows an example.

Our results consistently identify occupational gender biases previously documented in BERT-based language models. For instance, occupations historically dominated by one gender are clearly shown in decision tree visualizations to maintain strong gender associations. The interpretability afforded by the PAC-guaranteed decision trees not only confirms these biases but also clearly delineates affected occupations, providing actionable insights for bias mitigation.

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
