# OpenReview forum: "Extracting PAC Decision Trees from Black Box Binary Classifiers (Extended Abstract)"
_nesyconf.org/NeSy/2025/Conference_Phase_2 — NeSy 2025 - Phase 2 Poster_

### Official Review · Reviewer_rVaA · 2025-06-30
**Extended abstract: Extracting PAC Decision Trees from Black Box Binary Classifiers**

**Rating:** 7
**Confidence:** 4

**Review:**

In this paper, the authors discuss their contribution on learning decision trees freom binary black box classifiers under modified PAC estimates.  This is then applied to a BERT-based module to discover certain biases, which are then shown as rules.

I think the extended abstract does a good job of introducing the key concepts within the limited space. However, I do feel that some more depth could be added as there is still room left.

This was published at AAAI 2025. I think it is a good fit for NeSy given that it combine black box classifiers with symbolic decision trees.

**Anonymity:**

Remain anonymous

---

### Official Review · Reviewer_fcWi · 2025-07-08
**Extracting PAC Decision Trees from Black Box Binary Classifiers**

**Rating:** 8
**Confidence:** 4

**Review:**

This work presents a significant advancement in Explainable AI (XAI) by providing **formal theoretical guarantees for the fidelity of extracted decision trees** from black-box models.

**Core Contributions:**
*   **PAC-Guaranteed Fidelity:** The authors leverage the **Probably Approximately Correct (PAC) learning framework** to ensure that the extracted decision tree's predictions closely match the original neural network's behavior. This addresses a critical gap in existing XAI methods which lack such guarantees.
*   **Novel TrePAC Algorithm:** They introduce **TrePAC**, a new decision tree learning algorithm designed specifically for **binary classification tasks**. Unlike conventional methods, TrePAC uses the PAC framework to estimate necessary training data and tree complexity.
*   **Practical Realizability:** While retaining the realizability assumption of PAC learning, they innovatively allow the hypothesis to **misclassify 'k' examples** and still provide learnability bounds, making the approach more practical. A probabilistic definition of fidelity is also introduced.
*   **Model-Agnostic Explainability:** TrePAC functions as a **model-agnostic post-hoc explanation algorithm**, making it widely applicable to various pre-trained binary classifiers.

**Significance & Application:**
*   **Enhanced Trust and Interpretability:** By offering **provable guarantees of approximation accuracy**, the work significantly increases the trustworthiness and reliability of explanations, especially for sensitive predictions. The extracted decision trees are **inherently interpretable** and provide **explicit rules**.
*   **Bias Detection and Mitigation:** TrePAC was successfully applied to **BERT-based language models** to reveal and quantify **occupational gender biases**. The clear rules provided by the decision trees not only confirm these biases but also **delineate affected occupations**, offering **actionable insights for bias mitigation**.

**Pros:**
*   **Formal Fidelity Guarantees:** Provides **unprecedented theoretical guarantees** for surrogate model fidelity.
*   **High Interpretability:** Generates **inherently interpretable decision trees** with explicit, actionable rules.
*   **Effective Bias Unveiling:** Proven useful in identifying and quantifying complex biases in real-world models.
*   **Model-Agnostic:** Applicable to various black-box binary classifiers.

**Cons:**
*   **Binary Classification Focus:** The framework is explicitly focused on **binary classification problems**, potentially limiting its direct applicability to multi-class or more complex tasks without extensions.
*   **Partial Realizability Assumption:** While relaxed, the work *kept* the realizability assumption, which is still considered "a rather strong assumption in practice".
*   **Oracle Dependence:** TrePAC relies on a "membership oracle to classify examples used for training", the practical implications of which are not fully detailed in the provided abstract.

**Anonymity:**

Remain anonymous